# Dietary Acid-Base Balance in High-Performance Athletes

**DOI:** 10.3390/ijerph17155332

**Published:** 2020-07-24

**Authors:** Marius Baranauskas, Valerija Jablonskienė, Jonas Algis Abaravičius, Laimutė Samsonienė, Rimantas Stukas

**Affiliations:** 1Department of Physiology, Biochemistry, Microbiology and Laboratory Medicine of the Faculty of Medicine, Institute of Biomedical Sciences, Vilnius University, 01513 Vilnius, Lithuania; valerija.jablonskiene@mf.vu.lt (V.J.); algis.abaravicius@mf.vu.lt (J.A.A.); 2Department of Rehabilitation, Physical and Sports Medicine, Institute of Health Sciences of the Faculty of Medicine, Vilnius University, 01513 Vilnius, Lithuania; laimute.samsoniene@mf.vu.lt; 3Department of Public Health, Institute of Health Sciences of the Faculty of Medicine, Vilnius University, 01513 Vilnius, Lithuania; rimantas.stukas@mf.vu.lt

**Keywords:** high-performance athletes, actual nutrition, eating habits, diet, body composition, acid-base balance

## Abstract

Physical exercise leads to metabolic changes that affect the acid-base balance in skeletal muscles and other tissues. Nutrition is one of the factors that may influence the acid-base balance in the body. Keeping alkaline circumstances in the body is important not only for health and athletic performance in training but also during competition in many sport events. This is especially significant for athletes who practice in sport at the highest level of competition. The aim of the study was to determine the dietary acid-base balance in competitive Lithuanian high-performance athletes, and to evaluate the effect of actual diets of athletes on NEAP (net endogenous acid production), muscle mass and body mineral content during a four-year Olympic cycle. The research participants were 18.1 ± 3.3-year-old Lithuanian high performance athletes (*n* = 323). The actual diet was investigated using the 24 h recall dietary survey method. The measurements of body composition were performed using BIA (bioelectrical impedance analysis). The potential renal acid load of the diets of athletes (dietary PRAL) and NEAP were calculated. In 10.2% of athletes, NEAP exceeds 100 mEq · day^−1^ and is on average 126.1 ± 32.7 mEq · day^−1^. Higher NEAP in athletes is associated with lower muscle mass (β -1.2% of body weight, *p* < 0.001) but has no effect on the amount of minerals in the body (β 0.01% of body weight, *p* = 0.073). Overall, 25–30% of Lithuanian high-performance athletes use high-protein diets (2.0–4.8 g · kg^−1^
· day^−1^) leading to a dietary acid-base imbalance as well as an excessive production of endogenous acids in the body. Athletes are recommended to consume higher amounts of potassium and magnesium. An increase in calcium intake up to 1500 mg per day is recommended. In exceptional cases, periodised nutrition for athletes may involve diets complemented with bicarbonate and/or beta-alanine supplements.

## 1. Introduction

Acid-base balance homeostasis is essential for ensuring health and physical performance indicators. Organic acids are produced in the body during basal metabolism, while physical exercise can lead to additional acid production in the body [1]. When engaged in sports, even submaximal exercise induces metabolic changes that affect the acid-base balance in the skeletal muscles and other tissues [2]. Exercise intensity can lower blood pH from 7.4 to 6.9. It is noteworthy that the lowest blood pH reading (6.80–6.90) due to endogenous acids production was found in runners after a simulated 400 m race [3,4]. The increased H^+^ levels in myocytes during high-intensity exercise lead to acidosis and fatigue [1,5]. Muscle fatigue occurs due to H^+^ accumulation because the mitochondrial function and enzymatic activity are impaired. As a consequence, the production of glycolytic energy is disrupted [6,7]. It has also been proven that the concentration of H^+^ ions leads to the accumulation of interstitial K^+^, where proteins bind H^+^ ions instead of K^+^ ions. This causes the hyperpolarisation of cells, inhibits the rate of the nerve impulse propagation, triggers the changes in the membrane potential and, as a result, disrupts the muscle function [8]. HCO^3−^ in extracellular fluids is the major H^+^ buffer [9]. Therefore, the maintenance of a higher concentration of HCO^3−^ results in a faster removal of H^+^ from muscle cells [5]. Thus, an increase in the capacity of the acid buffer system improves the anaerobic [10,11] and aerobic [12] fitness of athletes. The maintenance of alkalinity in intracellular fluids enables a faster removal of H^+^ from muscle cells resulting in a delayed muscle fatigue which occurs due to increased acidosis [1].

Nutrition is one of the factors that can affect the acid-base balance in the body. This was confirmed by research that showed a strong relationship between the chemical composition of dietary intake and the urine pH range [13]. In this context, the possible influence of the intake of different foods on the potential renal acid load (PRAL) was assessed. The PRAL index that shows the potential renal acid load indicates the presence of milliequivalents (mEq) of H^+^ ions per 100 g of food. Most fruits and vegetables have a negative PRAL index because the biologically active substances found in them act as H^+^ buffers in the body. Meanwhile, foods high in protein and phosphorus have a positive PRAL index, which means that their consumption stimulates H^+^ production in the body.

According to the research, the dietary habits of the population in the developed countries with the typical Western diet are dominated by protein foods of animal origin (fish, meat, eggs), leading to high levels of metabolic acidosis in the body [14]. The similarity in problems such as high protein and fat intakes was found among athletes from many countries. Based on the research data reported in the scientific literature, it has been stated that a diet high in protein and fat, but low in carbohydrates was adopted by professional athletes from countries such as Poland [15,16], Iran [17], Kuwait [18], England [19,20], Brazil [21], Greece [22], Australia [23,24,25], France [26,27] Finland [28], China [29,30], Ireland [31], Netherlands [32], Spain [33,34], the United States of America (USA) [35], South Africa [36], Canada [37]. It has also been found that if athletes consume large amounts of protein and their diets are low in carbohydrates, they can suffer from metabolic acidosis which can adversely affect their physical performance [13,38,39,40]. In addition, insufficient consumption of potassium and magnesium with vegetables and fruits increases the risk of acidosis which may result in the reduced physical working capacity of athletes [41].

It should be noted that the presence of persistent acidosis in the body may trigger the impairment of the muscle function leading to the inhibition of muscle protein synthesis. Part of the amino acids from the degraded muscle proteins can be used for glutamine synthesis in the liver and, in later stages, for acid neutralisation. As a consequence, the increased acid production can lead to a decrease in muscle mass [42,43,44,45]. In addition, regular acidic diet may trigger a reduction in bone mineralisation, an increase in urinary calcium excretion and pose a risk of bone fractures [46].

There are no scientifically grounded data on how the diets of high-performance athletes impact their body’s acid-base balance, muscle mass and body mineral content. The aim of the study was to determine the dietary acid-base balance in competitive Lithuanian high-performance athletes, and to evaluate the effect of the actual diets of athletes on NEAP (net endogenous acid production), muscle mass and body mineral content during a four-year Olympic cycle.

## 2. Materials and Methods

### 2.1. Study Population

High-performance sport or elite sport is sport at the highest level of competition. The target population for the survey was high-performance athletes (*n* = 341) included in the lists, approved under the orders the National Olympic Committee of Lithuania. The main inclusion criteria for study participants was qualification standards that have been previously met by athletes.

Only those athletes that had already obtained an Olympic qualification quota place or the athletes who had participated in the European Athletics Championships and/or the World Athletics Championships for the purposes of Olympic qualification were investigated. Those athletes who had not participation in sports competitions on a professional level were excluded from the survey. The size of the sample group (*n* = 338) was selected using the OpenEpi Sample Size Calculator with a margin of error of 5% and probability of 99.9%. Over the period from 2017 through 2018, during a preparatory phase of training (macrocycle), 96% of the candidates (*n* = 323) to the Lithuanian Olympic team were included in study and investigated. The athletes ranged in age from 16 to 33 (the average mean age of the athletes was 18.1 ± 3.3 years) and were tested during the research and the training status of the athletes corresponded to 7.9 ± 3.8 years, while workouts were done 5.8 ± 0.8 days a week, with an average workout time of 175.6 ± 60.6 min a day. The dimensions of the training workload of athletes fully complied with the training plans approved by the Lithuanian Sports Centre and the National Olympic Committee of Lithuania. The training plans were specified in the Tokyo 2020 and PyeongChang 2018 programmes. The research sample included 72.4% (*n* = 234) men and 27.6% (*n* = 89) women. According to the dominant energy expenditure methods, the subjects were divided into anaerobic 40.2% (*n* = 130) and aerobic 59.8% (*n* = 193) fitness athletes [47]. The group of anaerobic athletes comprised weightlifters (*n* = 6), gymnasts (*n* = 3), discus, javelin throwers, shot put athletes (*n* = 6), jumpers (*n* = 4), basketball players (*n* = 52), boxers (*n* = 14), Greco-Roman wrestlers (*n* = 29), judo wrestlers (*n* = 12), and taekwondo wrestlers (*n* = 4). The group of athletes of aerobic fitness involved representatives of academic rowing (*n* = 36), road cyclists (*n* = 50), swimmers (*n* = 66), skiers (*n* = 17), biathletes (*n* = 20), long-distance runners (*n* = 13), representatives of modern pentathlon (*n* = 12) and representatives of figure skating (*n* = 2). A more detailed analysis of the study recruitment process and study procedures is provided in Figure 1.

### 2.2. Anthropometric Measures

The height measurements in athletes were taken at the Lithuanian Sports Medicine Centre using a stadiometer (± 0.01 m). The measurements of the body weight and the individual weight components (body weight (BW), lean body mass (LBM) (in kg and %), muscle mass (MM) (in kg and %), body fat (BF) (in kg and %) and mineral content in bones and electrolytes) (in kg and %) were performed at the Lithuanian Sports Centre using the bioelectrical impedance analysis (BIA) tetra-polar electrodes (13 lot 21 block with certification EN ISO (an international standard is adopted by the European Union) 13488; Jinryang Industrial Complex, Kyungsan City, South Korea) and resistivity was measured with 8–12 tangent electrodes at different frequencies of the signal: 5, 50, 250, 550 and 1000 kHz [47,48]. LBM, MM and mineral content were assessed according to the norms set for men and women. LBM norm for men is 75–85%, for women 70–80%; MM norm for men is 74–80%, for women 64–80%; mineral norm for men ranges between 5.8–6.0%, for women 5.5–6.0%. The muscle and fat mass index (MFMI) of each athlete was determined by dividing the weight of the muscle (in kg) by weight (in kg). The BF and the ratio of muscle and fat mass were evaluated according to the standards presented in Table 1 (MFMI) [47].

### 2.3. Energy Requirements

The basal metabolic rate (BMR), daily energy expenditure (DEE), training energy expenditure (TEE) were estimated in all the subjects. BMR was calculated using the Harris and Benedict formulas [49]. We collected 24-h records of physical activity on the same day when the participants recorded their dietary energy intake (EI). The physical activity levels and lifestyle variables (regular and non-regular activities, sedentary activities and sleeping habits) conform to the standards specified by the American Dietetic Association, Dietitians of Canada, and the American College of Sports Medicine [50]. These measures (the activity codes and metabolic equivalents (METs) (in kcal/kg/h) for physical activities) were supported by the studies of Ainsworth et al. [51] and the data were processed according to the specific activity.

### 2.4. Dietary Intake and Eating Habits

The 24-h actual nutrition survey method was employed to assess the actual nutrition in athletes [52,53,54]. The respondents were surveyed through the direct interview carried out by a specially trained interviewer at the Lithuanian Sports Centre. The actual nutrition survey method facilitated the compilation of the data on the amounts of food, meals, food supplements consumed by each athlete. To capture all foods and meals eaten, and their amounts, a special atlas of photos with different portions of foods and meals weighted in grams was used [55]. We evaluated the average daily food sets consumed by athletes on the basis of which the chemical composition and energy value of food rations were determined in line with the chemical composition tables [56]. The consumption of carbohydrates, proteins and fats was assessed taking into account the recommendations provided in the scientific literature [57,58]. The amount of carbohydrates recommended for athletes is 5–8 g · kg^−1^
· day^−1^, protein content is 1.4–2.0 g · kg^−1^
· day^−1^. The percentage of energy provided by fat should be between 20% and 35%. The daily intake of minerals and their compliance with the reference daily intake (RDI) was assessed according to the RDI of vitamins and minerals approved in Lithuania [59]. To study the eating habits, we designed and used a validated questionnaire originally constructed by M. Baranauskas [60]. The respondents participated in direct interviews. The questionnaire comprised questions about the socio-demographics (gender, age, place of residence, sport, sporting experience, etc.) and eating habits of athletes.

### 2.5. Potential Renal Acid Load (PRAL), Net Endogenous Acid Production (NEAP) and the Diets

The following formula was used to estimate the NEAP [61]: NEAP was estimated according to the equation (mEq · day^−1^) = PRAL^1^ (mEq · day^−1^) + OA^2^ (mEq · day^−1^) where PRAL shows the potential renal acid load of the estimated diet and OA (organic anions) shows the urinary organic anions under analysis, with the 2 components calculated as follows:

PRAL^1^ (mEq · day^−1^) = (0.49 × protein (g · day^−1^)) + (0.037 × phosphorus (mg · day^−1^)) − (0.021 · potassium (mg · day^−1^)) − (0.026 × magnesium (mg · day^−1^)) − (0.013 × calcium (mg · day^−1^)).

OA^2^ (mEq · day^−1^) = individual body surface area ^3^
× 41/1.73.

The body surface area was calculated according to the formula proposed by Du Bois and Du Bois [62]: ^3^ Individual body surface area (m^2^) = 0.007184 × (height (cm) ^0.725^
× weight (kg) ^0.425^.

### 2.6. Statistical Analysis

All the normally distributed continuous variables are presented as means ± standard deviations (SD), whereas the qualitative variables are presented as relative frequencies (in %). The normality of variable distribution was tested by the Shapiro–Wilk *W* test. When normality was confirmed, the *t*-tests of the independent samples were used to assess the differences observed between the groups. Pearson (*r*) correlation coefficient were used to determine the strength of the relationship between the variables under analysis. The correlation coefficient r can range in value from −1 to +1. A higher degree of the absolute value of the coefficient shows a stronger the relationship between the variables. The correlations above 0.4 are considered to be relatively strong; the correlations between 0.2 and 0.4 are moderate, and those below 0.2 are weak.

The multiple linear regression analysis was used to determine the association between the dietary intake and NEAP. The model was adjusted for gender and type of sport. Logarithmic or inverse square transformations were used to improve normality. By using a stepwise multivariate logistic regression method, we determined which eating habits of athletes depended on their sport. The stepwise multivariate logistic regression method was used to establish which eating habits determined PRAL ≤ 0 and PRAL > 0. The method of parameter estimation used in this study was maximum likelihood, and several techniques were employed to assess the appropriateness, adequacy and usefulness of the model using the likelihood-ratio test, Hosmer and Lemeshow (H-L) test statistic, Wald (W) statistic, and Nagelkerke R^2^ statistic. During the next stages, we calculated the logistic regression coefficients (β), odds ratios (OR) and their 95% confidence intervals (CIs) for each variable under analysis. All the reported *p*-values are based on two-sided tests and compared to a significance level of 5%. The statistical analysis was performed using Stata version 12.1 (StataCorp, College Station, TX, USA), SPSS V.25 for Windows (International Business Machines Corporation, Armonk, NY, USA) and Microsoft Excel (Microsoft Corporation, Redmond, WA, USA).

### 2.7. Ethics Statement

Prior to the research, all the organisational issues regarding the survey were discussed with the Lithuanian Sports Centre and the Bioethics Committee. The study was conducted in accordance with a permit to carry out biomedical research, issued by the Lithuanian Bioethics Committee (No. 158200-11-113-25, of 3 November 2009). Prior to testing, all the athletes provided a written consent and the study protocols were approved by the Institutional Review Board of the Lithuanian Sports Medicine Centre. The biomedical research was conducted according to the principles expressed in the Declaration of Helsinki.

## 3. Results

### 3.1. Characteristics of Respondents

The body composition (BW, LBM, BF MM, and MFMI) of athletes was examined as shown in Table 2. The height, BW, LBM, MM, and the mineral content (in bones and electrolytes) fluctuated within the norms. BF in male athletes (16.7 ± 4.7%) was acceptable (15–19%) while MFMI (5.2 ± 2.5) was high (4.7–6.0). Meanwhile, the BF and MFMI in female athletes differed in the groups of different sports. BF of female athletes involved in sports of anaerobic fitness was 24.9 ± 4.8%, which was acceptable (25–29%) and higher than BF of athletes of aerobic fitness which was 22.2 ± 3.6% and corresponded to the optimal FM (20–24%) (*p* = 0.005). In addition, MFMI (2.9 ± 0.8) in anaerobic female athletes corresponded to the low one (1.9–2.9). Meanwhile, a higher MFMI observed in aerobic women involved (3.4 ± 0.8, corresponding to the average of 3.0–3.9) confirms a more optimal body composition (*p* = 0.012).

### 3.2. Dietary Intake and Energy Expenditure

The examination of the actual diet of athletes revealed that the EI amounts to 3343 ± 1133 kcal and corresponds to DEE by 91.4 ± 27.8% (Table 2).

The evaluation of the nutrient intake showed that regardless of the type of sport, when training 175.6 ± 60.6 min per day, the amount of carbohydrates consumed (5.5 g · kg^−1^
· day^−1^) conforms to the minimum requirements (5–8 g · kg^−1^
· day^−1^). Nutrient imbalances in the diets of athletes are caused by an excessive fat intake. Irrespective of the type of sport, the share of energy value of fat in the diet of athletes (39.0 ± 7.8%) exceeds what is recommended (by 20–35%).

The average amount of protein of 1.7 ± 0.6 g · kg^−1^
· day^−1^ found in the diets of all types of athletes (anaerobic and aerobic) corresponds to what is recommended (1.4–2.0 g · kg^−1^
· day^−1^). The protein content recommended for athletes is no more than 2.0 g · kg^−1^
· day^−1^. However, according to our study, 29.2% of the athletes who develop anaerobic fitness consume 2.0–4.8 g · kg^−1^
· day^−1^ protein, and 24.4% of the athletes training for aerobic fitness consume 2.0–3.9 g · kg^−1^
· day^−1^ protein.

The consumption of phosphorus, potassium, magnesium and calcium by athletes exceed the recommended amounts. In contrast to aerobic athletes, anaerobic athletes consumed more phosphorus (*p* = 0.014), calcium (*p* = 0.022), and magnesium (*p* = 0.012). In terms of RDI, the anaerobic athletes consumed more phosphorus, calcium and magnesium by 2.9, 1.2 and 1.5 times respectively. Meanwhile, the amounts of phosphorus, calcium and magnesium in the diets of aerobic athletes exceeded RDI by 2.6, 1.4 and 1.4 times, respectively. Regardless of the sport, the amount of potassium consumed by the athletes was 1.5 times higher than RDI. In addition, the dietary amounts of calcium and phosphorus were unbalanced. This was confirmed by the calcium to phosphorus ratio (Ca/P) (0.6 ± 0.2) which was below the recommended 0.75 and resulted from the excessive dietary intake of phosphorus (Table 3).

### 3.3. Eating Habits and the PRAL

Lithuanian high-performance athletes rarely consume foods that should be found in their diets every day. The study revealed that 49.8% of athletes consumed bakery products, 29.6%—cereals, 43.7%—fresh vegetables, 43.7%—fresh fruits, and 36.8%—dairy products four to seven days a week. Dried fruits and boiled potatoes are consumed less frequently—25.9% and 40.1% of the athletes consumed dried fruit and boiled potatoes 2 days a week, 12.1% and 37.6% from 2 to 7 days a week, respectively.

In terms of the frequency of consumption of protein found in meat and fish products, 41.3% of the athletes chose poultry 2–4 days a week, while eggs (53.6%), beef (46.2%), pork (49%), fish (44.9%) and meat preparations (42.1%) were consumed less frequently, from 1 to 2 days a week.

The eating habits of athletes are likely to determine the potential renal acid load (PRAL) of their diets. After evaluating PRAL of athletes’ diets, it was found that more than half (65.9%) of the examined diets had a positive PRAL. PRAL (10.7 ± 42.1 mEq · day^−1^) of the diets of anaerobic athletes did not differ from PRAL (9.3 ± 35.9 mEq · day^−1^) of aerobic athletes and was also positive (PRAL > 0 mEq · day^−1^) (*p* = 0.761) (Table 3).

A stepwise multivariate logistic regression method was used to determine which eating habits of athletes determine the PRAL of their diets. Table 4 presents the OR estimating the association between the different food intakes by athletes and the dietary acid load among the participants who were identified with dietary PRAL ≤ 0 mEq · day^−1^. The final built logit model was tested with the Hosmer and Lemeshow goodness-of-fit test statistic (Nagelkerke R^2^ = 0.28; H-L stat χ^2^ = 14.5, *p* < 0.006). As indicated in Table 4, the probability of PRAL in the diets of athletes increased 1.4 times (OR 1.4) to become ≤ 0 mEq · day^−1^, when dairy products (*p* = 0.05), fresh vegetables (*p* = 0.048) and dried fruits (*p* = 0.046) were consumed more frequently. Specifically, milk and fresh vegetables were consumed more frequently by athletes in PRAL ≤ 0 group (47.7% and 52.3%, respectively) 4–7 days per week compared to athletes in PRAL > 0 group (31.1% and 39.1%, respectively). Similarly, more athletes (47.8%) with PRAL ≤ 0 consumed dried fruit more frequently (2–7 days a week) compared to the group of PRAL > 0 athletes (32.9%). In contrast, PRAL of the athletes who consumed more grain products was higher than 0 mEq · day^−1^ (OR 0.7, *p* = 0.050). Specifically, the athletes in PRAL > 0 group (71.4%) consumed grain products more frequently (2–7 days a week) compared to athletes in PRAL ≤ 0 group (62.6%).

### 3.4. Acid-Base Balance and Diets

Aiming to determine whether the chemical composition of the athlete diets was suitable for maintaining the body’s acid-base balance, the study assessed the effect of nutrition on the body’s net endogenous acid production (NEAP). It is important for athletes that NEAP did not exceed 100 mEq · day^−1^ for a longer period of time.

Although, according to our study, the average NEAP in anaerobic athletes (56.9 ± 43.3 mEq · day^−1^) did not differ from NEAP observed in aerobic athletes (53.7 ± 37.1 mEq · day^−1^) (*p* = 0.469), in 10.2% of Lithuanian high-performance athletes NEAP is higher than 100 mEq · day^−1^ and on average amounts to 126.1 ± 32.7 mEq · day^−1^.

After using a multivariate linear regression method, we found that with a 95% confidence level of the consumption of larger amounts of protein, phosphorus, and carbohydrates, NEAP increases from 34.1 to 167.2 mEq · day^−1^ (*p* < 0.001). Meanwhile, with the increased consumption of potassium, calcium, and magnesium with food, NEAP decreases from −9.1 to −100.4 mEq · day^−1^ (*p* < 0.05) (Table 5).

A more detailed analysis of the results showed that the daily protein intake (2.6 ± 0.8 and 2.4 ± 0.7 g · kg^−1^
· day^−1^) of anaerobic and aerobic athletes with NEAP > 100 mEq · day^−1^ exceeds the maximum recommended amounts by 1.3 times, and phosphorus content (2907.3 ± 892.2 and 2402.7 ± 516.3 mg · day-1)—by 4.1–3.4 times.

The correlation analysis also confirmed a moderate relationship between the protein intake and NEAP in the group of anaerobic athletes (r = 0.482, *p* < 0.001) and a weak relationship in the group of aerobic athletes (r = 0.274, *p* < 0.001) (Figure 2 and Figure 3).

The consumption of more protein foods, lower dietary PRAL and lower NEAP can be achieved by consuming sufficient amounts of the minerals—potassium, calcium and magnesium. According to the study, the consumption of potassium (4834.8 ± 1642.3 mg · day^−1^), calcium (1375.1 ± 562.3 mg · day^−1^) and magnesium (525.8 ± 150.8 mg · day^−1^) in athletes with NEAP > 100 mEq · day^−1^ did not significantly differ (*p* > 0.05) from those found in athletes with NEAP < 100 mEq · day^−1^, and exceeded RDI by 1.4, 1.5, and 1.5 times, respectively.

### 3.5. The Effect of Acid-Base Balance on the Muscle Mass and Body Mineral Content of Athletes

As indicated in Table 6, after applying the method of linear multivariate regression, it was found that the amount of muscle mass in athletes depended on the protein and phosphorus consumption, and the resulting NEAP in the body. At higher NEAP, the muscle mass (% of BW) was significantly lower by 1.2% (*p* < 0.001). However, only the excess phosphorus intake was associated with lower muscle mass (β −10.2% of BW, *p* < 0.001). Meanwhile, the athletes taking an increased amount of protein are characterized by an increase of 12.6% in the muscle mass of BW (*p* < 0.001).

The analysis of NEAP and the impact made on NEAP by nutrient components as well as their influence on the body mineral content (in bones and electrolytes) revealed no effects of NEAP (β −0.01%, *p* = 0.073). The results of our study showed that the body mineral content (% of BW) of athletes was increasing with higher protein (β 0.15%, *p* < 0.001) and calcium (β 0.06%, *p* = 0.04) consumption. Meanwhile, with higher amounts of phosphorus intake, lower amounts of body mineral content were observed (β −0.16%, *p* < 0.001) (Table 7).

## 4. Discussion

Daily high-intensity exercise causes stress to the body’s buffer systems. Even moderate-intensity exercise causes metabolic changes that affect the acid-base balance in skeletal muscles and other tissues. Intense exercise can lower blood pH from 7.4 to 6.9 in 1 min leading to very rapid muscle fatigue [1,3,4]. Another factor influencing the acid-base balance in the body is diet [13] which can lead to low-grade metabolic acidosis (MA) (arterial blood pH is close to 7.35) [63]. Low-grade MA is typical when NEAP reaches about 50 mEq · day^−1^. In other countries, Remer et al. [64] and Lemann [65] found that NEAP for young people was 40.1–50 mEq · day^−1^. Similar data were obtained in our study. The average NEAP for anaerobic athletes was 56.9 mEq · day^−1^, and that for aerobic athletes—53.7 mEq · day^−1^. However, as many as 10.2% of Lithuanian high-performance athletes had NEAP higher than 100 mEq · day^−1^ which averaged to 126.1 ± 32.7 mEq · day^−1^. Long-term NEAPs of 100–120 mEq · day^−1^ or more results in kidney overload with acid and thus a decreased availability of bicarbonates in the blood [66].

The research suggests that a high-protein, low-carbohydrate diet can lead to low-grade MA, high NEAP and have an adverse effect on physical performance [38,39,40]. We have obtained conflicting results confirming that athletes’ NEAP was driven by higher protein (β 34.1 mEq · day^−1^) and carbohydrate (β 21.5 mEq · day^−1^) intake. According to our study, the more frequent consumption of grain products by athletes acidified their dietary PRAL (OR 0.7). PRAL of concentrated carbohydrate grain products are positive due to their amino acids (PRAL 4.5–8.0) which determined the acid load [61]. Meanwhile, the impact of carbohydrate-containing products on higher NEAP, as identified by scientists, was based only on the consumption of fruit and vegetables [67]. Nonetheless, carbohydrate intake (5.5 g · kg^−1^
· day^−1^) among the athletes that we studied was relatively low for meeting the daily energy needs of 3600–3900 kcal having a 175-min workout [57]. The consumption of grain products, vegetables, fresh and dried fruits by the Lithuanian athletes is too low and infrequent. This can lead to insufficient levels of glycogen stores in the liver and muscles between sports practice sessions, an increased risk of overtraining, and a weakened immune system [68].

It should be emphasized that 25–30% of Lithuanian high-performance athletes consume 2.0–4.8 g · kg^−1^
· day^−1^ proteins per day, which exceeds the recommendations. There is still a debate as to whether an increased long-term protein intake among physically inactive, incapacitated people can impair their kidney function [69,70]. According to some studies, a long-term acidogenic diet combined with physical exertion results in the initial impairment of the renal function. The glomerular filtration rate has been shown to decrease with the diet of moderate PRAL for 12 weeks in physically active men and women [71]. Other data suggest that a long-term use of 2.5 to 3.5 g · kg^−1^
· day^−1^ protein did not impair the renal and hepatic functions in weightlifters [72,73,74,75].

Scientific studies have shown that low-grade MA caused by an excessive protein and phosphorus intake increases cortisol secretion (hypercortisolism), proteolysis and inhibits protein synthesis [76,77,78]. The rate of anabolism of muscle proteins is inversely proportional to the amount of acids present [79]. The results of our study confirmed that at higher NEAP, athletes had a significantly lower muscle mass, by 1.2% of BW. It is noteworthy that at low-grade MA, more muscle protein is broken down in order to neutralize dietary acids [80]. After the breakdown of muscle proteins, part of the amino acids released into the plasma is used for glutamine synthesis in the liver. Glutamine is further metabolised and degraded in the proximal renal tubules to alpha-ketoglutarate (AKG^2−^) and ammonium. AKG^2−^ metabolism to glucose (or O_2_ and CO_2_) consumes two H^+^ ions, which reduces the acid load. As a consequence, the use of amino acids for glutamine synthesis may result in a lack of amino acids for the synthesis of new proteins in the muscles and a decrease in muscle mass [80]. According to the data in our study, a higher protein intake leads to a higher muscle mass (β 12.6% of BW), has no association with muscle loss and is sufficient to ensure a positive nitrogen balance in athletes’ bodies. Higher protein intakes for athletes are essential to promote muscle hypertrophy during the workout process [81].

Theoretically, protein and phosphorus in diets may lead to an increased acid load and increased calcium excretion, which may increase the risk of osteoporosis [42]. We have not determined the effect of NEAP on the body mineral content of our subjects (β −0.01% of BW). In contrast, higher amounts of body minerals (in bones and electrolyte) depend on higher protein and calcium intakes (β 0.15 and 0.06 % of BW, respectively) and lower phosphorus consumption (β −0.16% of BW). Our results are consistent with the data obtained from other researchers suggesting that a high protein intake increases the intestinal calcium absorption, insulin-like growth factor-1 (IGF-1) concentration in the blood, and lowers parathyroid hormone levels. In this way, protein compensates for the negative protein-induced acid load on urinary calcium excretion [82]. To prevent osteoporosis, with an increased protein intake (2.0 g · kg^−1^
· day^−1^), higher amounts (>600 mg · day^−1^) of calcium are recommended [83]. Also, similar data from epidemiological studies suggest that high levels of phosphorus do not have adverse effects on bone metabolism when an adequate calcium quantity is consumed [84]. An adequate (1170.2 ± 538.3 mg · day^−1^) calcium intake is confirmed by the actual nutrition results of the athletes that we have studied. However, calcium and phosphorus in the diets of athletes are unbalanced (Ca/P 0.6; should be 0.75) due to an excessive phosphorus intake. Therefore, Lithuanian athletes are recommended to increase the calcium intake to 1500 mg · day^−1^ as recommended by the International Olympic Committee (IOC) [85], and to consume milk and dairy products more frequently.

The research has shown that the dietary intake to reduce NEAP requires adequate potassium and magnesium intakes with fruits and vegetables. Switching from a moderate-protein (1.3 g · kg^−1^ · day^−1^) diet to a high-protein (2.1 g · kg^−1^
· day^−1^) diet with a low intake of vegetables and fruits has been shown to significantly reduce blood pH and HCO^3−^ concentration. Therefore, high-performance athletes who are advised to consume increased protein (1.4–2.0 g · kg^−1^
· day^−1^) amounts are also advised to consume sufficient amounts of fresh vegetables and fruits (at least 400 g) [81]. In our study, lower potassium and magnesium intakes resulted in higher NEAP (β −100.4 and −39.2 mEq · day^−1^, respectively). On the other hand, our study showed that the athletes with high NEAP consumed the amount of potassium higher than RDI by 1.4 times. This suggests that potassium RDI in athletes is too low to ensure the neutral dietary PRAL. In addition, no tolerable upper intake level (UL) has been established [86,87]. In this regard, potassium levels recommended for athletes may be 2–3 times higher than RDI.

Thus, the eating habits of the athletes we have studied and those of the population of other countries are characterized by a high consumption of protein foods of animal origin, which is associated with low-grade MA [14]. In this case, MA can be normalized by changing the eating habits (ensuring low PRAL) [88] or by taking dietary supplements [89]. However, there are no scientific studies demonstrating the benefits of a short-term (4–9 days) vegetarian low-protein diet for aerobic physical working capacity. No change was observed in recording the increased respiratory exchange ratio (RER) in athletes due to an increase in “non-metabolic” CO_2_ during submaximal exercise [90]. Studies justifying the effects of low PRAL diets on RER relate only to long-term eating habits with a high intake of vegetables and fruits [1,91]. Additionally, studies have shown that high doses of sodium bicarbonate (150 mEq · day^−1^) neutralize food acids and minimize the total excretion of acids caused by ammonium. [14]. Thus, in exceptional cases, to increase the capacity of the buffer system while performing repeated bouts of exercise for 0.5–7 min, Lithuanian high-performance athletes are recommended to use sodium bicarbonate in doses of 0.30 g · kg^−1^
· day^−1^ 120–150 min before workouts [92]. To reduce intracellular acidification during 1–4 min repeated bout physical exercise, athletes are recommended to enrich their diets with beta-alanine supplements (65 mg · kg^−1^
· day^−1^) for 10–12 weeks [92,93]. On the other hand, food supplements with buffering characteristics cannot replace conventional foods that lower PRAL.

In summary, while athletes may require a higher protein intake, high-protein diets can promote metabolic changes due to the production of additional acids in the body and lead to very rapid muscle fatigue during exercise. Dietary acid-base balance is also important for variables such as skeletal muscle protein metabolism and bone mineralisation. According to our study results, an excessive production of endogenous acids in the body in athletes is associated with lower muscle mass and has no effect on the amount of minerals in the body. It is clear then that the interaction between the dietary acid-base balance and exercise in athletes needs to be further studied in order to better and more accurately assess the contribution of alkaline diet in athletic performance and the variables like the rate of protein synthesis and the breakdown and bone density. Therefore, further research is needed to assess the impact of higher fruit and vegetable consumption by athletes on the indicators of their physical performance between workouts.

Actual nutrition data combined with objective readings of body composition of athletes allow us to predict and implement targeted measures and recommendations for optimising the athlete nutrition for the next Olympic cycle. The data and recommendations of our study can be applied in practice by including them in the current sports training programmes like Tokyo 2020 and Beijing 2022. In the future, continuous investigation and monitoring of body composition and actual nutrition should be carried out during each 4-year Olympic cycle.

However, in the course of our study, we did not include selection criteria such as comorbidities, because professional athletes in Lithuania do not have long-term health-related clinical symptoms or their combinations and they are completely healthy. The athletes’ health monitoring is carried out every three months at the Lithuanian Sports Medicine Center. The health professionals ensure good health indicators of athletes in Lithuania. If serious health problems are identified during the monitoring process, the athlete is officially prohibited from exercising and participating in any level of competition. Therefore, the limitations of our study are related to the fact that while conducting our study we did not add the inclusion and exclusion criteria of this study such as the economic status, health indicators of the athletes (e.g., exercise-related injuries, iron deficiency anaemia, short-term renal impairments due to rapid bodyweight reduction among wrestlers and/or boxers). These variables may have associations with the actual diet or eating habits of athletes and these are the directions for further research. Another limitation of our study is that it was only a 24-h dietary recall survey of actual nutrition during the pre-competition period. Thus, in the future, in cooperation with the Lithuanian Sports Medicine Center (LSMC), it is necessary to monitor the actual nutrition and other health indicators of high-performance athletes for a period of three to seven days during the preparatory and competition periods.

## 5. Conclusions

Regardless of the type of sport, the diets of Lithuanian high-performance athletes do not meet requirements. The diets of athletes are too high in fat. Even 25–30% of athletes practice high protein (2.0–4.8 g · kg^−1^
· day^−1^) diets which result in dietary acid-base imbalance as well as excessive production of endogenous acids in the body. Higher NEAP in athletes is associated with lower muscle mass and has no effect on the body mineral content.

The training of athletes in different sports needs to be very individualised in terms of mesocycling of sports goals, including changes in the body composition, giving priority to the formation of eating habits. The eating habits of athletes need to be changed and carefully planned in practice to ensure acid-alkaline balance in the body by consuming alkaline-producing foods. In order to ensure the optimal dietary acid-base balance and to maximize muscle adaptation to exercise, athletes are recommended to consume higher amounts of potassium and magnesium found in fresh vegetables and dried fruits—twice as much as RDI. A more frequent consumption of milk and dairy products is recommended in order to increase calcium intake to 1500 mg per day. In exceptional cases, the diets of athletes should be enriched with bicarbonate and/or beta-alanine supplements.

## Figures and Tables

**Figure 1 ijerph-17-05332-f001:**
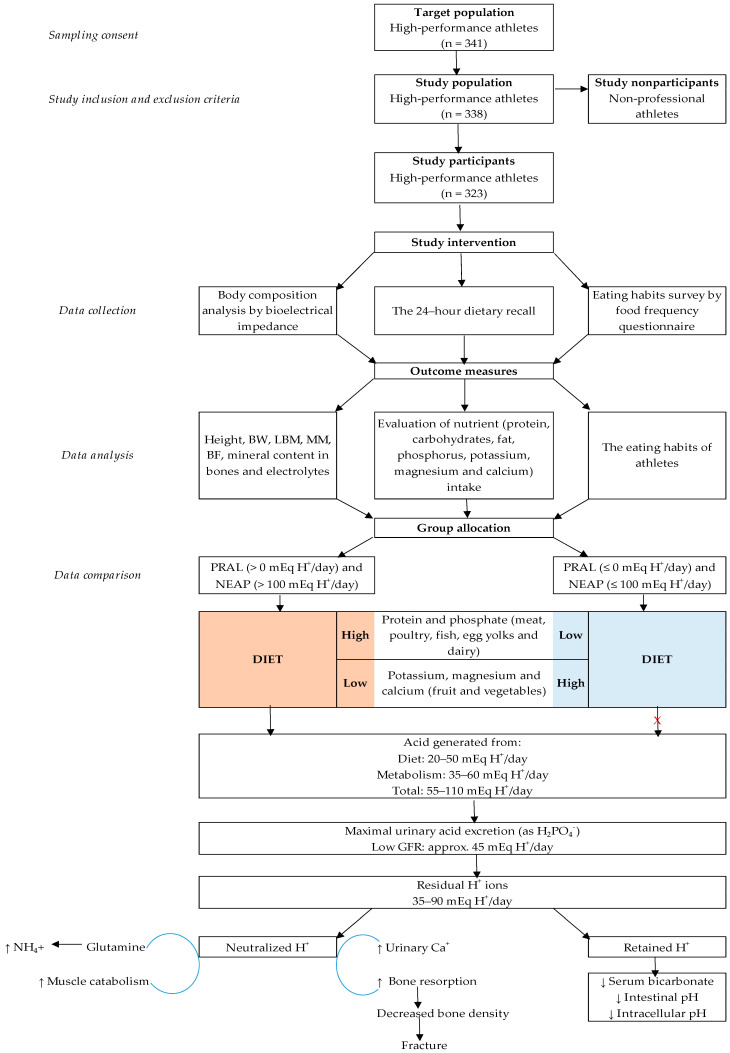
Flowchart of the enrollment of athletes and study procedures. BW—body weight; LBM—lean body mass; MM—muscle mass; BF—body fat; PRAL—potential renal acid load; NEAP—net endogenous acid production; GFR—glomerular filtration rate.

**Figure 2 ijerph-17-05332-f002:**
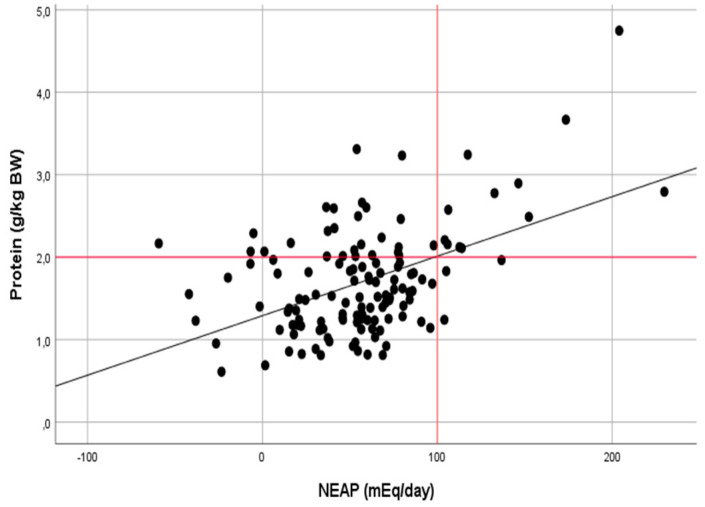
Relationship between the dietary protein intake (g · kg^−1^
· day^−1^) and NEAP (mEq · day^−1^) in athletes of anaerobic sports (r = 0.482, *p* < 0.001).

**Figure 3 ijerph-17-05332-f003:**
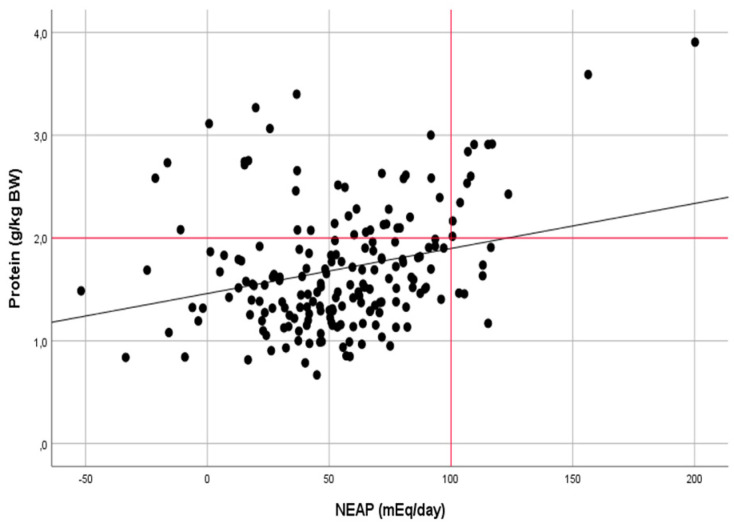
Relationship between the dietary protein intake (g · kg^−1^
· day^−1^) and NEAP (mEq · day^−1^) in athletes of aerobic sports (r = 0.274, *p* < 0.001).

**Table 1 ijerph-17-05332-t001:** Body fat (BF) percentage and muscle and fat mass index (MFMI) scale for athletes (by gender).

BF			MFMI		
Value	Males	Females	Value	Male Athletes	Female Athletes
Too low	<5%	<15%	Insufficient	<2	<1.8
Lean	5–9%	15–19%	Too small	2.1–3.39	1.9–2.89
Optimal	10–14%	20–24%	Moderate	3.4–4.69	3–3.99
Acceptable	15–19%	25–29%	Extensive	4.7–6.0	4–5
Excessive	20–24%	30–34%	Maximum	>6	>5

BF—body fat; MFMI—muscle and fat mass index.

**Table 2 ijerph-17-05332-t002:** Body composition of athletes (by sport and gender).

Body Composition	Anaerobic Sports		Aerobic Sports	
	Male	Female	Male	Female
Height (m)	1.83 ± 0.15	1.73 ± 0.11	1.83 ± 0.08	1.67 ± 0.06
BW (kg)	77.5 ± 17.4	67.4 ± 14.3	75.0 ± 11.6	59.7 ± 7.5
LBM (kg)	63.8 ± 11.4	50.1 ± 7.9	62.2 ± 7.7	46.3 ± 4.7
LBM (% of BW)	83.3 ± 5.3	75.2 ± 4.9	83.3 ± 4.2	77.6 ± 3.7
MM (kg)	59.3 ± 10.5	46.2 ± 7.2	57.9 ± 7.0	42.8 ± 4.3
MM (% of BW)	77.4 ± 5.2	69.4 ± 4.8	77.6 ± 4.1	72.1 ± 3.6
MFMI	5.3 ± 2.4	2.9 ± 0.8	5.2 ± 2.6	3.4 ± 0.8
BF (kg)	13.7 ± 7.1	17.6 ± 7.1	12.9 ± 4.7	13.4 ± 3.5
BF (% of BW)	16.7 ± 5.3	24.9 ± 4.8	16.7 ± 4.2	22.2 ± 3.6
Minerals (kg) ^1^	4.5 ± 1.0	3.9 ± 0.8	4.4 ± 0.7	3.4 ± 0.4
Minerals (% of BW) ^1^	5.8 ± 0.1	5.8 ± 0.1	5.8 ± 0.1	5.8 ± 0.1

BW—body weight; LBM—lean body mass; MM—muscle mass; MFMI—muscle and fat mass index; BF—body fat; ^1^—mineral composed of bone and electrolyte. The data is normally distributed and presented as means ± standard deviation (SD).

**Table 3 ijerph-17-05332-t003:** Dietary intake of athletes.

Nutrition Profile	Anaerobic Sports ^1^	Aerobic Sports ^2^	*t*-Test^1/2^	
	Mean ± SD		t	*p*
DEE (kcal · day^−1^)	3894 ± 876	3595 ± 864	3.032	0.003
EI (kcal · day^−1^)	3457 ± 1280	3266 ± 1020	1.486	0.138
EI (kcal · kg^−1^ · day^−1^)	47 ± 16	47 ± 15	−0.230	0.818
CHO (g · kg^−1^ · day^−1^)	5.4 ± 2.0	5.5 ± 2.2	−0.268	0.789
CHO (%)	46.5 ± 7.3	46.3 ± 8.9	0.206	0.837
PRO (g · kg^−1^ · day^−1^)	1.7 ± 0.6	1.7 ± 0.6	0.084	0.933
FAT (%)	38.9 ± 7.1	39.1 ± 8.2	−0.270	0.788
K (mg · day^−1^)	5189.7 ± 2341.8	4790.6 ± 1984.5	1.647	0.100
Ca (mg · day^−1^)	1254.0 ± 580.2	1113.9 ± 501.9	2.310	0.022
Mg (mg · day^−1^)	522.7 ± 240.7	461.6 ± 191.0	2.534	0.012
P (mg · day^−1^)	1999.0 ± 788.1	1806.9 ± 600.1	2.483	0.014
Ca/P ratio	0.6 ± 0.2	0.6 ± 0.2	0.680	0.497
PRAL (mEq · day^−1^)	10.7 ± 42.1	9.3 ± 35.9	0.305	0.761
NEAP (mEq · day^−1^)	56.9 ± 43.3	53.7 ± 37.1	0.725	0.469

The values are expressed as mean ± SD; EI—energy intake; DEE—daily energy expenditure; BW—body weight; PRO—protein; CHO—carbohydrate; FAT—fat; Ca—calcium; P—phosphorus; Mg—magnesium; K—potassium; PRAL—potential renal acid load; NEAP—net endogenous acid production. Significant differences set by independent samples Student’s *t*-test between groups: ^1^—group 1, ^2^—group 2.

**Table 4 ijerph-17-05332-t004:** Effects of athletes’ eating habits on their dietary PRAL.

PRAL ≤ 0 (mEq · day^−1^) ^a^	β	SE	W	*p*	Exp (β) (95% CI)
Grain products	−0.3	0.2	3.5	0.050	0.7 (0.5; 1,1)
Dairy products	0.3	0.2	3.7	0.050	1.4 (1.0; 2.0)
Fresh vegetables	0.3	0.2	3.3	0.048	1.4 (1.0; 2.0)
Dried fruits	0.3	0.2	3.6	0.046	1.4 (1.0; 2.0)
Constant	−2.2	0.8	7.6	0.006	0

^a^—reference category is PRAL > 0 mEq · day^−1^; β—is the estimated coefficient, with standard error SE (<5); W is the Wald test statistic; Nagelkerke R^2^ = 0.28; Exp (β) is the predicted change in odds for a unit increase in the predictor (odds ratio (OR)); CI—confidence interval. The final model was tested with the Hosmer and Lemeshow goodness-of-fit test statistic (H-L stat χ^2^ = 14.5, *p* < 0.006).

**Table 5 ijerph-17-05332-t005:** Effects of carbohydrates, proteins, phosphorous, potassium, calcium, magnesium consumed by athletes on their net endogenous acid production (NEAP).

NEAP (mEq · day^−1^)	β	95% CI	*p*
PRO (g · kg^−1^ · day^−1^) (ln)	34.1	(23.0; 45,1)	<0.001
CHO (g · kg^−1^ · day^−1^) (ln)	21,5	(11.0, 31.8)	<0.001
P (mg · day^−1^) (ln)	167.2	(152.4; 181.9)	<0.001
K (mg · day^−1^) (ln)	−100.4	(−108.3; −92.6)	<0.001
Ca (mg · day^−1^) (ln)	−39.2	(−45.6; −32.8)	<0.001
Mg (mg · day^−1^) (ln)	−9.1	(−17.9; −0.2)	0.044
EI (kcal · kg^−1^ · day^−1^) (ln)	−69.2	(−85.6; −52.8)	<0.001

The influence of dietary intake on NEAP (mEq · day^−1^) is estimated controlling for athlete sport and gender (adjusted for sports type and gender). F (9, 313) = 201.2, *p* < 0.0001, R^2^ = 0.85. PRO—protein; CHO—carbohydrate; Ca—calcium; P—phosphorus; Mg—magnesium; K—potassium; EI—energy intake; CI—confidence interval.

**Table 6 ijerph-17-05332-t006:** Effects of athletes’ NEAP, protein and phosphorus consumption on their muscle mass (% of BW).

Muscle Mass (% of BW)	β	95% CI	*p*
NEAP (mEq · day^−1^) (ln)	−1.2	(−1.8; −0.7)	<0.001
PRO (g · kg^−1^ · day^−1^) (ln)	12.6	(10.8; 14.5)	<0.001
P (mg · day^−1^) (ln)	−10.2	(−12.1; −8.2)	<0.001

Muscle mass (% of body weight) is estimated controlling for athlete sport and gender (adjusted for sports type and gender). F (5, 295) = 78.1, *p* < 0.0001, R^2^ = 0.56. BW—body weight; NEAP—net endogenous acid production; PRO—protein; P—phosphorus; CI—confidence interval.

**Table 7 ijerph-17-05332-t007:** Effects of athletes’ NEAP, protein, phosphorus, and calcium consumption on their body mineral content (% of BW).

Body Mineral (% of BW) ^1^	β	95% CI	*p*
NEAP (mEq · day^−1^) (ln)	−0.01	(−0.03; −0.001)	0.073
PRO (g · kg^−1^ · day^−1^) (ln)	0.15	(0.10; 0.19)	<0.001
P (mg · day^−1^) (ln)	−0.16	(−0.23; −0.09)	<0.001
Ca (mg · day^−1^) (ln)	0.06	(0.02; 0.09)	0.004

Body mineral content (% of body weight) is estimated controlling for athlete sport and gender (adjusted for sports type and gender). F (6, 294) = 12.6, *p* < 0.0001, R^2^ = 0.20. NEAP—net endogenous acid production; PRO—protein; P—phosphorus; Ca—calcium; CI—confidence interval. ^1^—minerals in bones and electrolytes.

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
