# Peer review of "Dietary Acid-Base Balance in High-Performance Athletes"

_ijerph, 2020, doi:10.3390/ijerph17155332_

Round 1
Reviewer 1 Report
The manuscript describes dietary acid balance in aerobic and anaerobic type young athletes and evaluates acidicity (NEAP) of the diets.
The manuscript is not ready for publication.
GENERAL:
- in the results section there should be only your own results NOT comparing using other results. The place for them is in discussion.
- discussion is too long...say exactly the most important discussions and conclusions
- English language should be improved by an expert (too much using e.g. s-genetive)
- high protein in athletes e.g. 1.4-2.0 is important, but it increases acidicity. So the other part of nutrition must be very carefully planned and done in practice. This must be emphasised more in conclusion and in abstract. To keep alkaline circumtances in body is very important for health and athletic performance in training and in competition in many sport events.
SPECIFIC REMARKS:
- in the title "high performance" is questionable" because n=323 and they are young (18.1 y)...use only "young athletes". In methods you can tell that they were Lithunian athletes.
- page 61: not low levels but high levels of metabolic acidosis...( low pH but high acidosis)
- in introduction it should be told that maximum pH (lowest) is found in 400m runners (45-50s duration) 6.80-6.90 (classic Kindermann and Keul 1977: Anaerobe energiebereitstellung im hochleistungssport, and e.g. Vilmi et al 2016, 5,3, Journal of Athletic Enhancement)
- METHODS: page 80: the sentence is poor: On average, the investigations..something to do with training?? The athletes are not high-performance! They are young national athletes on an average.
- is 24 hours too short time period for gathering data?
- RESULTS: do not show other results ...only your own in this paragraph. Body fat numers are suprisingly high: in males around 16% and in femalaes around 22-25%. Not high performance athletes!
- too much decimals (e.g. 3894.6 kcal no decimals) e.g. height 183.2 cm better 1.83 m)
- DISCUSSION: is too long in scientific presentation/sense. 318: pH goes under 7.00 in 50 second maximal performance see Vilmi et al 2016. How to write references in the text in this journal ? page 321: authors Remer T and Lemann J found...??
Author Response
Responses to the observations provided by the reviewers
Observations by Reviewer 1
Reviewer’s observation: „In the results section there should be only your own results NOT comparing using other results. The place for them is in discussion. RESULTS: do not show other results ...only your own in this paragraph. “
Reply and corrections. We have taken into account the reviewer‘s observation and deleted the cited literature source and comparative information.
Reviewer’s observation: „English language should be improved by an expert (too much using e.g. s-genetive)“
Reply and corrections. We have corrected the English language according to the given observations.
Reviewer’s observation: ”High protein in athletes e.g. 1.4-2.0 is important, but it increases acidicity. So the other part of nutrition must be very carefully planned and done in practice. This must be emphasised more in conclusion and in abstract. To keep alkaline circumtances in body is very important for health and athletic performance in training and in competition in many sport events.“
Reply and corrections. All the observations were taken into consideration, and the sections of Abstract and Conclusions were supplemented with the following information: A) “Nutrition is one of the factors that may influence the acid-base balance in the body. To keep alkaline circumstances in the body is very important for health and athletic performance in training and during competition in many sport events.” B) “The training of athletes in different sports needs to be very individualised in terms of mesocycling of sports goals, including the changes in the body composition, giving priority to the formation of eating habits. The eating habits of athletes need to be changed and carefully planned in practice to ensure the acid-alkaline balance in the body by consuming alkaline producing foods. In order to ensure the optimal dietary acid-base balance and to maximize muscle adaptation to exercise, athletes are recommended to consume higher amounts of potassium and magnesium found in fresh vegetables and dried fruits - twice as much as RDI.”
Reviewer’s observation: “In the title "high performance" is questionable" because n=323 and they are young (18.1 y)...use only "young athletes". In methods you can tell that they were Lithunian athletes. METHODS: page 80: the sentence is poor: On average, the investigations..something to do with training?? The athletes are not high-performance! They are young national athletes on an average.
Reply and corrections. There is no specific age limit for taking part in the Olympic Games. Therefore, taking into account the reviewer’s observation, we have clarified the information provided in the methodology section of the manuscript: High performance sport or elite sport is sport at the highest level of competition. The target population for the survey was high-performance athletes (N=341) included in the lists, approved under the orders the National Olympic Committee of Lithuania. The main inclusion criteria for study participants were qualification standards that have been previously met by athletes. Only those athletes who had already obtained an Olympic qualification quota place or the athletes who had participated in the European Athletics Championships and/or the World Athletics Championships for the purposes of Olympic qualification were investigated. The athletes ranged in age from 16 to 33 (the average mean age of the athletes was 18.1 ± 3.3 years) were tested during the research and the training status of the athletes corresponded to 7.9 ± 3.8 years, while workouts were done 5.8 ± 0.8 days a week, with an average workout time of 175.6 ± 60.6 min a day. The dimensions of the training workload of athletes fully complied with the training plans approved by the Lithuanian Sports Centre and the National Olympic Committee of Lithuania. Training plans were specified in the Tokyo 2020 and PyeongChang 2018 programmes.
Reviewer’s observation: „page 61: not low levels but high levels of metabolic acidosis...( low pH but high acidosis)“
Reply and corrections. Following the reviewer‘s observation, we have clarified the information as follows: “According to the research, the dietary habits of the population in the developed countries with the typical Western diet are dominated by protein foods of animal origin (fish, meat, eggs), leading to high levels of metabolic acidosis in the body.”
Reviewer’s observation: „In introduction it should be told that maximum pH (lowest) is found in 400m runners (45-50s duration) 6.80-6.90 (classic Kindermann and Keul 1977: Anaerobe energiebereitstellung im hochleistungssport, and e.g. Vilmi et al 2016, 5,3, Journal of Athletic Enhancement)“
Reply and corrections. Following the observation, the Introduction was supplemented accordingly: When engaged in sports, even submaximal exercise induces metabolic changes that affect the acid-base balance in the skeletal muscles and other tissues. In connection with that, maximum blood pH (6.80-6.90) is found in runners after the simulated 400 m race.”
Reviewer’s observation: “Is 24 hours too short time period for gathering data? Body fat numers are suprisingly high: in males around 16% and in femalaes around 22-25%. Not high performance athletes!“
Reply and corrections. In the article, we additionally indicated specific references to literature sources in which the research methods used in other studies were confirmed (24-food recall, portion sizes, BIA, food habits questionnaire). Accordingly, this research method is completely sufficient when the entire population of high-performance athletes in Lithuania was studied. The body composition of athletes was evaluated using the bioelectrical impedance analysis (BIA) tetra-polar electrodes and resistivity was measured with 8–12 tangent electrodes at different frequencies of the signal: 5, 50, 250, 550 and 1000 kHz. The equipment itself scans quite deep layers of body tissues. Therefore, the data are relatively accurate. In addition, other researchers and authors use other methods of examining body composition. A gold standard for body composition testing has not been established yet. Also, the results of different measurement methods cannot be compared with each other. We compared the data with the norms for especially high-skilled athletes, which are published by sports researchers and scientists in Lithuania. We have indicated these standards in the manuscript methodology section. In Lithuania, we additionally calculate the ratio of muscle and fat mass of highly skilled athletes, which helps to optimize the body composition of athletes by special means, so that athletes in a particular sport achieve the best possible results in sports.
Reviewer’s observation: “Too much decimals (e.g. 3894.6 kcal no decimals) e.g. height 183.2 cm better 1.83 m)“
Reply and corrections. The manuscript has been revised accordingly.
Reviewer’s observation: “DISCUSSION: is too long in scientific presentation/sense; Discussion is too long...say exactly the most important discussions and conclusions“
Reply and corrections. Following the observation, we have shortened the discussion section. In the discussion section, we have refused the excess discussion information on PRAL, dietary habits, dairy and calcium consumption, the effects of vegetarian diet on anerobic and aerobic performance, the effects of bicarbonates and beta-alanine supplements (leaving only recommendations). In the Discussion section we left only the essential results that have a significant science relationship to the key variables.
Reviewer’s observation: “318: pH goes under 7.00 in 50 second maximal performance see Vilmi et al 2016.“
Reply and corrections. Following the observation, we have clarified the data: Intense exercise can lower blood pH from 7.4 to 6.9 in 1-minute leading to very rapid muscle fatigue.
Reviewer’s observation: “How to write references in the text in this journal ? page 321: authors Remer T and Lemann J found...??“
Reply and corrections. We have corrected to citation of the authors, folowing the observations and recommendations: In other countries, Remer et al. […] and Lemann […] found that NEAP for young people was 40.1–50mEq day-1.

Reviewer 2 Report
Thank you for giving me the opportunity to review the article. The authors conducted a study on the dietary acid-base balance in high-performance athletes in Lithuania. The topic may be important to think about health maintenance among athletes. However, there are several methodological concerns. I listed the comments for further consideration below.
Comments:
Abstract:
- The authors should add the definition of the high-performance athletes in the Abstract briefly.
Introduction and Background:
- The authors should describe the dietary habits and/or related studies focusing on athletes in detail.
Materials and Methods:
- The definition of “Lithuanian high-performance athletes” should be clearly stated.
- The authors should add the recruitment process of the study participants.
- The authors should add the inclusion and exclusion criteria of this study.
- Why did the authors not collect the economic status (and other related variables) of the study participants?
- Did all the study participants have no diseases? The health status of the study participants can affect the dietary habits.
- The category of the “strength of the relationship” should be added in the Statistical Analysis section.
- The authors should add the methods for calculating the sample size.
- The authors conducted the study between 2017 to 2018, but the ethical approval was obtained in 2009. Why the time between the ethical approval and the study is so long?
Results:
- According to the recruitment process of this study, the authors should add a flow diagram of this study.
- The authors presented the results base on the multivariate logistic regression. The reviewer recommends the authors to show the parameters of model fitting.
Discussion:
- The generalizability (and implications for the potential readers in foreign countries) of this study should be added.
- The authors should add the sentences of the limitations of this study.
Author Response
Observations by Reviewer 2
Reviewer’s observation: “The authors should add the definition of the high-performance athletes in the Abstract briefly.“
Reply and corrections. Taking into account the reviewer‘s observation, we corrected the Abstract as follows: “To keep alkaline circumstances in the body is important for health and athletic performance in training and during competition in many sport events. This is especially significant for athletes who practice of sport at the highest level of competition. The aim of the study was to determine the dietary acid-base balance in competitive Lithuanian high-performance athletes, and to evaluate the effect of the actual diets of athletes on NEAP (net endogenous acid production), muscle mass and body mineral content during four-year Olympic cycle.” We have also made changes to the Introduction and clarified the objective of the research.”
Reviewer’s observation: “The authors should describe the dietary habits and/or related studies focusing on athletes in detail.“
Reply and corrections. Following the reviewer’s recommendation, we examined the scientific literature and provided data on the actual nutrition of athletes from other countries and identified the key nutrition issues that are common to many athletes. The Introduction section was supplemented as follows: “According to the research, the dietary habits of the population in the developed countries with the typical Western diet are dominated by protein foods of animal origin (fish, meat, eggs), leading to high levels of metabolic acidosis in the body. The similarity in problems such as high protein and fat intakes was found among athletes from many countries. Based on the research data reported in the scientific literature, it has been stated that a diet high in protein and fat, but low in carbohydrates was adopted by professional athletes from the countries such as Poland, Iran, Honk Kong, Kuwait, England, Brazil, Greece, Australia, France, Finland, China, Ireland, Netherlands, Spain, JAV, South Africa, Canada It has also been found that if athletes consume large amounts of protein and their diets are low in carbohydrates, they can suffer from metabolic acidosis which can adversely affect their physical performance.”
Reviewer’s observation: “The definition of “Lithuanian high-performance athletes” should be clearly stated. The authors should add the recruitment process of the study participants. The authors should add the methods for calculating the sample size. The authors should add the inclusion and exclusion criteria of this study.”
Reply and corrections. After taking into account the reviewer‘s observations we upgraded the section of Methodology: “High performance sport or elite sport is sport at the highest level of competition. The target population for the survey was high-performance athletes (N=341) included in the lists, approved under the orders National Olympic Committee of Lithuania. The main inclusion criteria for study participants were qualification standards that had been previously met by athletes. Only those athletes that had already obtained an Olympic qualification quota place or the athletes who participated in the European Athletics Championships and/or the World Athletics Championships for the purposes of the Olympic qualification were investigated. Those athletes who had not participated in sports competitions on a professional level were excluded from the survey. The size of the sample group (N = 338) was selected using the OpenEpi Sample Size Calculator with the error of 5% and with the probability of 99.9%. Over the period from 2017 through 2018, during a preparatory phase of training (macrocycle), 96% of the candidates (N=323) to the Lithuanian Olympic team were included in the study and investigated. The athletes ranging in age from 16 to 33 (the average mean age of the athletes was 18.1 ± 3.3 years) were tested during the research and the training status of the athletes corresponded to 7.9 ± 3.8 years, while workouts were done 5.8 ± 0.8 days a week, with an average workout time of 175.6 ± 60.6 min a day. The dimensions of the training workload of athletes fully complied with the training plans approved by the Lithuanian Sports Centre and the National Olympic Committee of Lithuania. Training plans were specified in the Rio 2016 and PyeongChang 2018 programmes.”
Reviewer’s observation: „Why did the authors not collect the economic status (and other related variables) of the study participants? Did all the study participants have no diseases? The health status of the study participants can affect the dietary habits.“
Reply and corrections. The aim of our study was very specific to determine the influence of the diet of athletes on acid and alkali balance. We assessed the impact of eating habits on PRAL. We also purposefully assessed the relationships between NEAP and the body composition of athletes (muscle mass and mineral content in the body), which is quite relevant and new in the current scientific context. Of course, the diet of athletes, economic status, health indicators (e.g., exercise-related injuries, iron deficiency anaemia) may be interrelated, and these are the directions for further research. Due to the abundance of the data and purposefulness of the research, we did not use this data in our current specific study.
Reviewer’s observation: “The category of the “strength of the relationship” should be added in the Statistical Analysis section.“
Reply and corrections. Following the observations, we have upgraded the section of Statistical Analysis: “The correlation coefficient can range in value from −1 to +1. The larger the absolute value of the coefficient shows the stronger the relationship between the variables. Correlations above 0.4 to be relatively strong; correlations between 0.2 and 0.4 are moderate, and those below 0.2 are considered weak.”
Reviewer’s observation: „The authors conducted the study between 2017 to 2018, but the ethical approval was obtained in 2009. Why the time between the ethical approval and the study is so long?“
Reply and corrections. In 2009, the general permit was issued by the Bioethics Committee to the Institute of Public Health of Vilnius University to perform biomedical research in general (not for one specifically), therefore it was used to carry out the current research. During the intersectoral cooperation of the Faculty of Medicine of Vilnius University with the Lithuanian Sports Medicine Centre (LSMC), the investigation of athletes was mutually coordinated. The evaluation of the body composition of athletes and their nutrition was part of an overall health study.
Reviewer’s observation: „According to the recruitment process of this study, the authors should add a flow diagram of this study.“
Reply and corrections. Following the reviewer‘s observations, we have upgraded the methodology of the research. For accuracy, we provided the flowchart of the enrollment of athletes and study procedures in the Study Population included in the Materials and Methods”.
Reviewer’s observation: „The authors presented the results base on the multivariate logistic regression. The reviewer recommends the authors to show the parameters of model fitting.“
Reply and corrections. The notes below Table 4 include the fitting parameters of the logit model. However, for the sake of accuracy, according to the reviewer's recommendations, we have clarified the description of the results and the text, too: „Table 4 presents the OR estimating the association between the different food intakes by athletes and the dietary acid load among the participants who were identified with dietary PRAL ≤ 0 mEq day-1. The final builded logit model was tested with the Hosmer and Lemeshow goodness-of-fit test statistic (Nagelkerke R2 = 0.28; H-L stat χ2 = 14.5, p < 0.006).”
Reviewer’s observation: „The generalizability (and implications for the potential readers in foreign countries) of this study should be added.“
Reply and corrections. After taking into account the reviewer‘s observations, we have generalised the research data targeting at the potential readers in foreign countries: “In summary, while athletes may require a higher protein intake, high protein diets can promote to metabolic changes due to the additional acids production in the body and lead to very rapid muscle fatigue during exercise. Dietary acid-base balance is also important for variables such as skeletal muscle protein metabolism and bone mineralization. According to our study results, an excessive production of endogenous acids in the body in athletes is associated with lower muscle mass and has no effect on the amount of minerals in the body. It is clear then that the interaction between athlete’s dietary acid-base balance and exercise needs to be further studied in order to better and more accurately assess the contribution of the alkaline diet in athletic performance and variables like the rate of protein synthesis and breakdown and bone density. Therefore, further research is needed to assess the impact of higher amounts of fruit and vegetables on the indicators of athletes’ physical performance between workouts.”
Reviewer’s observation: „The authors should add the sentences of the limitations of this study.“
Reply and corrections. We have corrected the section of Discussion accordingly as follows: “The limitation of this study is that it was only a 24 h food recall survey of actual nutrition during the pre-competition period. Therefore, in the future, in cooperation with the Lithuanian Sports Medicine Centre (LSMC), it is necessary to monitor the actual nutrition and other health indicators of high-performance athletes for a period from three to seven days during the preparatory and competition periods.”

Round 2
Reviewer 1 Report
The authors have made good improvements.
Author Response
We appreciate your prompt review of the article. After taking all the comments provided by the reviewers into consideration, we are submitting a revised article.
Together with the co-authors, we have essentially revised the manuscript.
Responses to the observations provided by the reviewers
Observations by Reviewer 1
Reviewer’s observation and corrections: We appreciate your professional review of our article, what resulted in a better quality of the article.
Respectfully
Co-authors of the article

Reviewer 2 Report
Thank you for giving me the opportunity to review the revised version of this article. The authors corrected the manuscript according to the comments mostly. However, an additional revision should be needed before acceptance for publication. I listed the comments for further consideration below.
AR, authors’ response; AC, additional comment
Materials and Methods:
- The authors should add the inclusion and exclusion criteria of this study.
AR: The main inclusion criteria for study participants were qualification standards that had been previously met by athletes. Only those athletes that had already obtained an Olympic qualification quota place or the athletes who participated in the European Athletics Championships and/or the World Athletics Championships for the purposes of the Olympic qualification were investigated. Those athletes who had not participated in sports competitions on a professional level were excluded from the survey.
AC: Why did the authors not set a criterion of comorbidities? These athletes can have comorbidities, it should affect the results of this study. If the authors did not collect the information, it is a limitation of this study which should be discussed.
- Why did the authors not collect the economic status (and other related variables) of the study participants?
AR: The aim of our study was very specific to determine the influence of the diet of athletes on acid and alkali balance. We assessed the impact of eating habits on PRAL. We also purposefully assessed the relationships between NEAP and the body composition of athletes (muscle mass and mineral content in the body), which is quite relevant and new in the current scientific context. Of course, the diet of athletes, economic status, health indicators (e.g., exercise-related injuries, iron deficiency anemia) may be interrelated, and these are the directions for further research. Due to the abundance of the data and purposefulness of the research, we did not use this data in our current specific study.
AC: The authors should add a limitation which they mentioned in the response in the Discussion section.
Author Response
We appreciate your prompt review of the article. After taking all the comments provided by the reviewers into consideration, we are submitting a revised article.
Together with the co-authors, we have essentially revised the manuscript.
Responses to the observations provided by the reviewers
Observations by Reviewer 2
Reviewer’s observations: A) “The authors should add the inclusion and exclusion criteria of this study. Why did the authors not set a criterion of comorbidities? These athletes can have comorbidities; it should affect the results of this study. If the authors did not collect the information, it is a limitation of this study which should be discussed.”; B) “Why did the authors not collect the economic status (and other related variables) of the study participants? The authors should add a limitation which they mentioned in the response in the Discussion section.”
Reply and corrections. In the course of our study, we did not include the selection criteria such as comorbidities, because professional athletes in Lithuania do not have long-term health-related clinical symptoms or their combinations, they are completely healthy. The athletes’ health monitoring is carried out every three months at the Lithuanian Sports Medicine Center. The health professionals ensure good health indicators of athletes in Lithuania. If serious health problems are identified during the monitoring process, the athlete is officially prohibited from exercising and participating in any level of competition. In this case, the athlete is removed from the preliminary and/or Olympic shift lists of athletes. In conducting the study, we followed the lists of athletes prepared by the Lithuanian National Olympic Committee. The athletes with health problems were not included in these lists, so this selection criterion was not needed. We are grateful to the reviewer for the observations. Also, in order to increase the qualitative value of the manuscript, we took into account the reviewer's observation and supplemented the Discussion section with the following information:
“However, in the course of our study, we did not include the selection criteria such as comorbidities, because professional athletes in Lithuania do not have long-term health-related clinical symptoms or their combinations, they are completely healthy. The athletes’ health monitoring is carried out every three months at the Lithuanian Sports Medicine Center. The health professionals ensure good health indicators of athletes in Lithuania. If serious health problems are identified during the monitoring process, the athlete is officially prohibited from exercising and participating in any level of competition. Therefore, the limitations of our study are related to the fact that while conducting our study we did not add the inclusion and exclusion criteria of this study such as the economic status, health indicators of the athletes (e.g. exercise-related injuries, iron deficiency anaemia, short term renal impairments due to rapid bodyweight reduction among wrestlers and/or boxers). These variables may have associations with the actual diet or eating habits of athletes and these are the directions for further research. Another limitation of our study is that it was only a 24-h dietary recall survey of actual nutrition during the pre-competition period. Thus, in the future, in cooperation with the Lithuanian Sports Medicine Center (LSMC), it is necessary to monitor the actual nutrition and other health indicators of high-performance athletes for a period of three to seven days during the preparatory and competition periods.”
Respectfully
Co-authors of the article
